# Three-Dimensional Dendritic Au–Ag Substrate for On-Site SERS Detection of Trace Molecules in Liquid Phase

**DOI:** 10.3390/nano12122002

**Published:** 2022-06-10

**Authors:** Yunpeng Shao, Sha Li, Yue Niu, Zezhou Wang, Kai Zhang, Linyu Mei, Yaowu Hao

**Affiliations:** 1School of Mechanical Engineering, North University of China, Taiyuan 030051, China; shypeng@nuc.edu.cn (Y.S.); ls950907@163.com (S.L.); ny057613@163.com (Y.N.); wzz4436@163.com (Z.W.); zhangk950417@gmail.com (K.Z.); 2Department of Materials Science and Engineering, University of Texas at Arlington, Arlington, TX 76019, USA

**Keywords:** 3D dendritic Au–Ag substrate, SERS, FDTD method, on-site detection, liquids

## Abstract

The development of a facile surface-enhanced Raman scattering (SERS) sensor for the on-site detection of trace molecules in liquid phase is a compelling need. In this paper, a three-dimensional (3D) dendritic Au–Ag nanostructure was constructed by a two-step electro displacement reaction in a capillary tube for the on-site liquid phase detection of trace molecules. The multiplasmon resonance mechanism of the dendritic Au–Ag structure was simulated using the finite-difference time domain (FDTD) method. It was confirmed that the highly branched 3D structure promoted the formation of high-density “hot spots” and interacted with the gold nanoparticles at the dendrite tip, gap, and surface to maximize the spatial electric field, which allowed for high signal intensification to be observed. More importantly, the unique structure of the capillary made it possible to achieve the on-site detection of trace molecules in liquids. Using Rhodamine 6G (R6G) solution as a model molecule, the 3D dendritic Au–Ag substrate exhibited a high detection sensitivity (10^−13^ mol/L). Furthermore, the developed sensor was applied to the detection of antibacterial agents, ciprofloxacin (CIP), with clear Raman characteristic peaks observed even at concentrations as low as 10^−9^ mol/L. The results demonstrated that the 3D dendritic Au–Ag sensor could successfully realize the rapid on-site SERS detection of trace molecules in liquids, providing a promising platform for ultrasensitive and on-site liquid sample analysis.

## 1. Introduction

Surface-enhanced Raman scattering (SERS) has been emerging as a powerful spectroscopic technique due to its extremely high sensitivity, fingerprint character and rapid nondestructive detection of molecules at an ultralow concentration [1,2,3,4,5]. In general, local surface plasmon resonance preferentially occurs in the nanoscale gaps between metal nanostructures [6,7,8]. Due to the improved signal feedback generated by the electromagnetic enhancement effect and practical requirements, SERS technology exploration further focuses on how to construct the active substrate with excellent performance and optimize its application. Compared with a two-dimensional planar substrate, 3D dendritic structures with highly branched tips can provide more abundant SERS active sites under the combined action of tip coupling and cumulative induction and become SERS sensors with great potential [9,10,11,12]. However, the traditional SERS detection process usually requires soaking a sample solution or dropping a sample solution on the surface of substrate, which makes it difficult to avoid sample contamination, poor signal uniformity and other limitations. The fluidity of the sample solution will also shorten the contact time between probe molecules and the SERS substrate, resulting in the analyte having a lower absorption ability and insufficient detection sensitivity [13,14,15]. Based on the above facts, the application of a 3D substrate to the rapid detection of liquid trace molecules is limited. There is an urgent need to develop a new analytical system with high sensitivity, high selectivity and real-time detection performance.

Among the reported emerging technologies, SERS substrates based on capillaries are expected to become ideal platforms for field microanalysis due to their advantages of low dose, high efficiency, precise process control and portability [16,17,18,19]. Since Carboni synthesized silver nanoparticles in channels using a chemical method, an SERS sensor formed by injecting or integrating active nanoparticles into a microchannel allowed for a simple, in situ analysis process [20]. However, the uncontrollable mixing time and inherent memory effect of nanocolloids and sample molecules would greatly reduce the detection’s efficiency and repeatability. To explore and optimize the performance of SERS devices, the in situ preparation of high-dimensional solid substrate became a better way to integrate SERS technology with on-site detection [21,22,23,24]. The complex and ordered solid substrate produces more resonance hotspots on different surfaces and provides additional features such as analyte capture and impurity filtration, making the detection process more reproducible, efficient and environmentally friendly. Some recent publications have presented methods for the in situ fabrication of microchannel-SERS devices. Chi et al. [25] proposed a method for the in situ polymerization of a 3D network porous silica material decorated with gold nanoparticles (3D-PSM@AuNP) in capillary tubes, and the new technique had demonstrated the ability to detect 4-mercaptopyridine in aqueous solution. Rajasimha [26] used a femtosecond laser sintering method to integrate silver nanoparticles onto a fused silica-based channel, forming a uniform, repeatable and large-area microchannel-SERS substrate. However, there are still few relevant practices and little theoretical research on the in situ preparation of high-performance SERS devices. It is inevitable that microchannel-SERS devices are used to further study their construction strategies and test methods to improve sample flux and realize ultrasensitive, on-site detection.

Here, we propose a manufacturing technology for a 3D Au–Ag substrate SERS sensor based on capillaries, forming an efficient, on-site, rapid-detection platform. The sensor was formed by replacing metal copper wire in the capillary tube, which combines the excellent chemical stability of Au and the enhancement performance of Ag’s high Raman signal. The spatial structure and morphology of the sensor were changed to a large extent, showing better physical and chemical properties. The detection sensitivity was greatly improved using the strong electromagnetic enhancement between Au/Ag particles and the nanoparticle gaps in dendrites. The unique capillary structure effectively controlled the flow time and space and enabled the aggregation of high-density “hot spots”, which essentially enhanced the interaction between the target molecule and sensor substrate, making it more suitable for the on-site detection of low-concentration samples. In addition, combined with simulation analysis, the influence of a 3D dendritic Au–Ag substrate structure on SERS sensor performance is systematically studied, and the optimal reaction parameters are obtained. Due to these characteristics, the optimized device can easily realize the on-site detection of a liquid medium, which provides a strong theoretical basis and experimental guidance for expanding the application scope of SERS detection.

## 2. Materials and Methods

### 2.1. Chemicals and Characterization

Analytical-grade silver nitrate (AgNO_3_, >99%) and Rhodamine 6G (R6G, 99%) were purchased from Alfa Aesar. Ciprofloxacin (CIP, 98%) and chloroauric acid (HAuCl4, >99%) was purchased from Macklin. Copper wires were purchased from Guan Tai Metal Materials Co., LTD. Water used throughout these experiments was purified with a Millipore system.

The morphologies and structures of Au–Ag particles were characterized using scanning electron microscopy (SEM) and energy-dispersive spectrometry (EDS, JSM-7900F). The crystalline structure was characterized by X-ray powder diffraction with a Cu Ka source (Siemens D800, Munich, Germany, λ = 0.154 nm) scanning from 20° to 90° at the rate of 2° per minute. The corresponding working voltage and current were 40 kV and 25 mA, respectively. The chemical purity of synthesized 3D Au–Ag substrate was studied by X-ray photoelectron spectroscopy (XPS).

### 2.2. Fabrication of Capillary-Based 3D Dendritic Ag Substrate

According to Figure 1, a 0.7 mm square capillary glass tube was used as the reaction channel. Three 0.05 mm pure-metal copper wires were ultrasonically cleaned for 10 min and then inserted into the capillary tube. Both ends of the glass tube were sealed, leaving only liquid flow channels. When the reaction occurred, 3 mL of 0.01 mol/L AgNO_3_ solution was injected into the reaction channel at a constant rate through the injection pump and reacted with copper wire for 15 min. After the reaction, ultrapure water was injected to clean the residual silver nitrate solution and silver nanoparticles suspended in the channel were cleaned. Finally, 3D dendritic silver SERS substrate was obtained between the surface of copper wire and the inner wall of channel pipe.

### 2.3. Fabrication of Capillary-Based 3D Dendritic Ag Substrate

Using the 3D silver dendrite prepared above as the basic unit, 0.005 mol/L chloroauric acid solutions of different volumes (1–5 mL) were injected into the reaction channel at a uniform rate until many black materials were formed on the surface of channel. Optimally, the injection pump speed was adjusted to 0.2 mL/min, and the HAuCl_4_ solution was continuously injected for 15 min. After the reaction, ultrapure water was injected at uniform speed to clean the residual chloroauric acid solution and suspended gold nanoparticles, and an SERS-active Au–Ag capillary sensor was successfully prepared.

### 2.4. Finite-Difference Time Domain (FDTD) Simulation

The finite-difference time domain (FDTD) method was used to simulate and analyze the spatial electric field distribution of a dendritic Au–Ag structure excited by a 532 nm laser. Based on the experimental parameters, the corresponding 3D model was constructed. Combined with the experimental data, the influence mechanisms of branch spacing, branch order and structural components on the SERS performance of dendritic substrate were explored. Furthermore, the particle size of gold nanoparticles was adjusted to analyze the multiple plasma coupling effects between gold nanoparticles and silver dendrite to form a complete theoretical model of electromagnetic field enhancement mechanisms.

### 2.5. SERS Measurement

A series of different solution concentrations were continuously delivered into the substrate channel at the injection pump flow rate of 0.2 mL/min and continued through the substrate for 60 min. The channel was then repeatedly cleaned with deionized water to remove residual probe molecules. After drying at room temperature to avoid light, SERS spectra were recorded on a model confocal microscopy Raman system (Horiba Raman Spectrometer, Kyoto, Japan) to study the SERS effect of substrates. The excitation wavelength was 532 nm. The integral time was 10 s, and the laser power was 5 mW, using a 100× objective with a laser spot of 1 μm in diameter.

## 3. Results and Discussion

### 3.1. Optical and Physical Characterization

Figure 2a is a typical SEM diagram, showing the reaction between silver dendrites and chloroauric acid solution. It can be observed that the layered fractal dendrites formed by diffusion restriction growth have large surface areas and abundant branch tips. EDS measurements in Figure 2c,f also clearly show that the initial silver in the sample is gradually replaced by gold nanoparticles, with a gold content of up to 39%, confirming that the nanostructure prepared by the substitution reaction is composed of Au–Ag bimetal. The subsequent XRD patterns in Figure 2b also confirm this result. The sample peaks can be labelled as (111), (200), (220), (311) and (222) diffraction peaks in the face-centered cubic (FCC) crystal Au–Ag alloy. The XRD patterns are basically the same as those of the original silver dendrites because the unit cell size of the bimetallic alloy changes by less than 1%. It is noteworthy that the intensity ratio of the bimetallic diffraction peak (111)/(220) is higher than that of the standard document, confirming that Au–Ag nanoparticles are more abundant on the (111) plane and preferentially grow in the direction parallel to (111).

The chemical purity of the nanostructure synthesized with a reaction time of 15 min was confirmed from the XPS results, as given in Figure 2g. The typical Au–Ag characteristic peak was observed after the denudate Au–Ag structure was calibrated with charge (referring to the C1s standard peak of 284.8 eV), which proves the successful loading of gold. The peaks corresponding to Ag 3d and Au 4f are presented with high-resolution scans in panel (h) and (i) of Figure 2, respectively. The positions of Ag 3d_5/2_ and Au 4f_7/2_ peaks are very close to the ones for metallic Ag and Au (368 eV and 84 eV, respectively). Thus, the XPS results confirm that while synthesizing Au–Ag nanostructures, both Au and Ag are in the metallic state.

### 3.2. Effect of Reaction Time on 3D Au–Ag Substrate

Figure 3 shows the SEM images of the morphology evolution of Au–Ag substrate over different reaction times. After the 5 min displacement reaction, the overall sample morphology still showed the initial dendritic structure, and a small number of tiny gold nanoparticles were formed on the surface, gap and tip of the dendrite, as shown in Figure 3a,b. With the increase in reaction time, Ag atoms were oxidized, leading to the gradual consumption of Ag dendrites, and a high number of gold nanoparticles were deposited on the dendrite surface. The initial edge branches rapidly evolved into rod or spherical branches, resulting in the gap becoming smaller, the tip becoming dull and the branch structure disappearing. However, after a long displacement reaction time (30 min), a high number of silver particles were consumed and gradually replaced by gold nanoparticles. The dendritic structure struggled to maintain its morphology, forming foliated, rod-shaped and granular structures, and many holes appeared on the surface (Figure 3e,f).

### 3.3. FDTD Calculations

Based on the experimental results, a corresponding asymmetric model of branches was established. The diameter of the main branches was 240 nm, the diameter of lateral branches was 160 nm, the diameter of secondary branches was 100 nm, and the distances between adjacent branches were 80 nm and 40 nm, respectively (Figure 4a,b). Uniform gold nanoparticles with a particle size of 60 nm were constructed at the tip, surface and gap of branches in the left part of trunk (Figure 4c,d). By comparison, the presence of gold nanoparticles made the electric field enhancement stronger than the results of silver dendrite simulation, but the overall enhancement trend was consistent with the previously studied results of silver dendrite simulation [27]. The electric field intensity reached the maximum in the trunk-lateral-secondary branch structure with a 40 nm spacing in adjacent branches. The results indicated that the enhancement of electromagnetic field could not solely be attributed to the plasmon resonance coupling between adjacent branches and dense sub-branches. The synergy between gold nanoparticles, gold nanoparticles and the whole-branch structure, silver dendrite branches and silver nanoparticles maximized the enhancement effect. It was believed that this structure could produce more sensitive SERS detection results. Furthermore, the sizes of gold particles were adjusted to 90 nm and 120 nm to explore the influence of the particle size of gold nanoparticles on the spatial field intensity (Figure 4e,f). It was found that, with the increase in particle size, electromagnetic field intensity would slightly decrease. This was because the metal sphere surface with a large particle size was smooth and the section surface was nearly flat, so the light field did not effectively excite dipole plasma. When the size was too low, the electrical conductivity of metal nanoparticles would sharply decrease due to electron scattering, and the field strength efficiency would decrease accordingly. However, due to the various limitations in our actual experiment, the particle size of gold nanoparticles deposited on the surface of silver dendrite was kept at about 90 nm.

### 3.4. SERS Performance

To verify the Raman enhancement effect of the substrate, R6G with a concentration of 10^−9^ mol/L was used as the probe molecule to obtain the Raman spectra of different enhanced substrates. Figure 5a shows that the main Raman peaks in R6G appeared at 1356, 1501 and 1645 cm^−1^, related to the stretching vibration of the aromatic group. The vibration bands at 606, 764 and 1176 cm^−1^ were related to the vibration in the C-C-C plane, the bending out of the C-H plane and the bending in the C-H plane, respectively. The Raman peaks at 1304 and 1567 cm^−1^ belonged to the bending vibration mode in the N-H plane (shown in Appendix A) [28,29]. In order to further quantify the amplification factor of the 3D SERS substrate signal, the analytical enhancement factor (AEF) of SERS substrate was calculated by the following equation:(1)$$\mathrm{AEF}=\frac{I_{SERS}}{C_{Raman}}/\frac{I_{SERS}}{C_{Raman}}$$
where $I_{SERS}$ and $I_{Raman}$ were the peak intensities in an SERS spectrum and a normal Raman spectrum, respectively. $C_{SERS}$ and $C_{Raman}$ were the analyte concentrations in the SERS measurement and the normal Raman measurement, respectively. Based on the strength at 1645 cm^−1^, the AEFs of Ag and Au–Ag were 8.21 × 10^7^ and 3.64 × 10^8^, respectively. By comparing its signal intensity, we found that the Raman enhancement of the Au–Ag substrate was much greater than that of pure silver dendrite. This SERS enhancement might be because the bimetal substrate had a large surface area and pores on the surface. When the sample solution flowed through the substrate, it presented periodic oscillations of a three-vortex structure, which was conducive to increasing the enrichment capacity of analytes. At the same time, the plasma coupling resonance effect not only existed between the particles in the dendrite structure but also existed in the vertical direction between the Au/Ag particles, which essentially enhanced SERS activity and provided the sensor with high sensitivity and a fast response. In addition, we selected an aqueous solution with a concentration of 10^−9^ mol/L R6G as the target molecule to study the SERS activity of Au–Ag substrate prepared under different reaction times. The results (Figure 5b) show that when the reaction time is 15 min, the signal intensity of the substrate increases by nearly one time, which can be used as the optimal solution for the subsequent detection process.

The practical application of Raman spectrum analysis depends on the SERS sensitivity of the substrate structure. To determine the detection limit of substrate, the dendritic Au–Ag microfluidic SERS device prepared under optimal conditions was used for the in situ detection of R6G. SERS spectra of R6G solutions with different concentrations (10^−9^–10^−13^ mol/L) were collected with a factor of 532 nm and integration time of 10 s. The Raman spectrum showed that, with the decrease in R6G aqueous solution concentration, the Raman intensity gradually weakened, and the LOD values of R6G could be achieved at 10^−13^ mol/L (Figure 6a). The intensity of Raman characteristic peak had a good linear relationship with the logarithmic concentration of R6G solution, and the best linear relationship was found at 1645 cm^−1^. The results showed that the device could meet the quantitative detection requirements for trace R6G. Furthermore, the signal variation recorded from 20 randomly selected sites was less than 15%, indicating the good reproducibility of hot spots on Au–Ag bimetallic structures.

### 3.5. Application for Detection of Ciprofloxacin

The microfluidic SERS device was used as an in situ detection platform for trace ciprofloxacin solution. The ciprofloxacin structure (shown in Figure 7) contained different characteristic functional groups, among the other carboxyl, amino, hydroxyl, cyclopropane, benzene, pyridine and amide groups. With the optimized dendritic Au–Ag nanostructure as the active substrate, the Raman characteristic peak in ciprofloxacin solution was significantly enhanced. The LOD values of ciprofloxacin could be achieved at 10^−9^ mol/L, and clear Raman characteristic peaks still were observed at the lowest concentration, which was better than the results reported in the literature. Based on the literature, the Raman spectral vibration peaks in ciprofloxacin were assigned and identified [30,31]. For example, the characteristic peak in the 635–695 cm^−1^ range was due to the out-of-plane vibrations of the ring. The peak at 735 cm^−1^ was considered to be a methylene swing pattern. Bands 1023 cm^−1^ and 1354 cm^−1^ were attributed to C-H oscillation and pyrazine ring mixed vibrations, respectively. The strongest Raman band at 1390 cm^−1^ could be attributed to O-C-O symmetric tensile vibration. The bands near 1484, 1552 and 1628 cm^−1^ were caused by the asymmetric stretching vibration of the benzene ring, the stretching vibration of the quinolone ring system and the asymmetric stretching vibration of aromatic ring C=O (shown in Appendix A). This was slightly different from the peak positions obtained in the literature, which might have been caused by the different enhancement effects of different active substrates on the functional groups in ciprofloxacin molecules, and the bonding of functional groups or other adsorption processes would affect the electron cloud distribution of ciprofloxacin molecules, thus presenting as a slight shift in the Raman peaks.

## 4. Conclusions

In summary, based on typical displacement reactions, a 3D dendritic Au–Ag structure was successfully constructed in a capillary tube and used as an effective sensor for liquid trace molecules. The nucleation and growth process of Au–Ag alloy could easily be controlled by adjusting the reaction time and an optimized 3D dendritic nanostructure could be obtained. The tips and clearances of a highly branched substrate endowed the sensor with a higher density of hotspots and showed the best SERS-enhanced effect. The FDTD simulation study also confirmed this point. The solution to be tested flowed through the substrate, and its large surface area and 3D notch efficiently captured target molecules, which proved the ultrasensitive detection ability of the R6G solution (10^−13^ mol/L). In addition, the SERS sensor had a high sensitivity for the on-site detection of ciprofloxacin, with a detection limit of 10^−9^ mol/L, which was expected to achieve the rapid and in situ analysis of various liquid pollutants.

## Figures and Tables

**Figure 1 nanomaterials-12-02002-f001:**
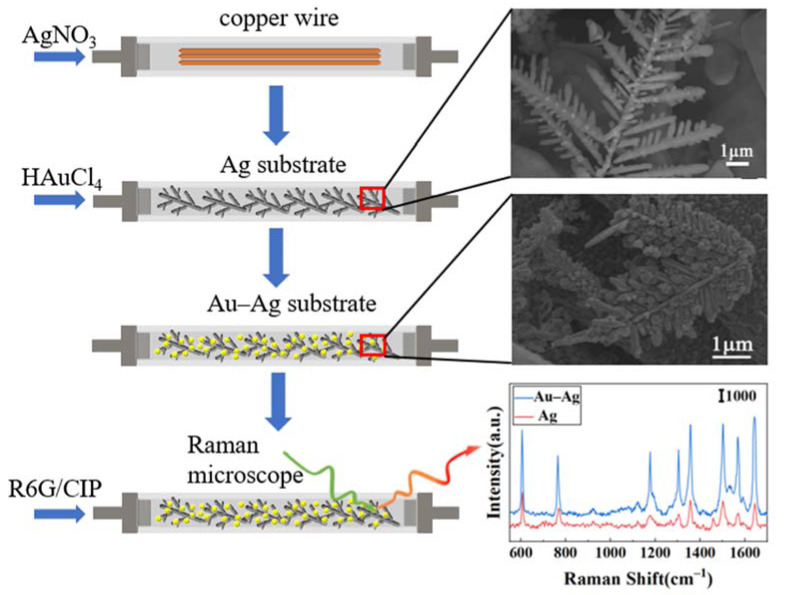
Schematic diagram of capillary-based 3D dendritic Au–Ag substrate.

**Figure 2 nanomaterials-12-02002-f002:**
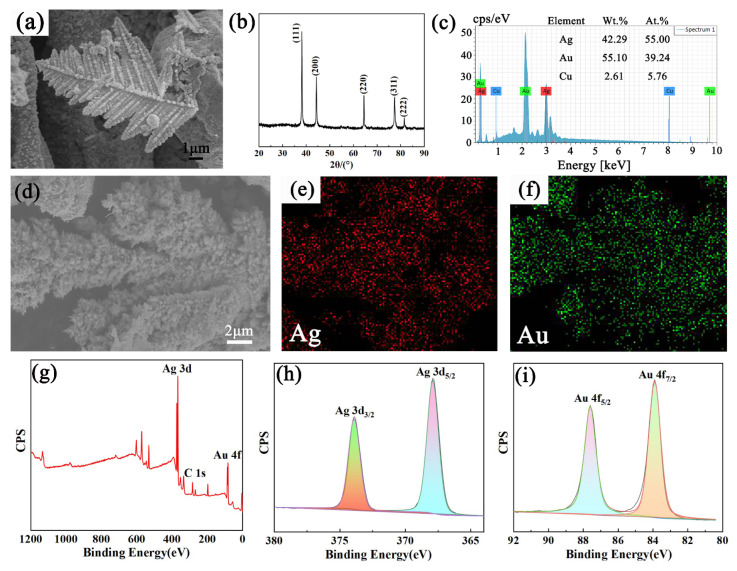
The dendritic Au–Ag structure was obtained by reaction with 3 mL chloroauric acid solution for 15 min. (**a**) SEM, (**b**) XRD, (**c**) EDS, (**d**) SEM, (**e**,**f**) EDS mapping, (**g**) XPS and (**h**,**i**) high-resolution XPS spectra for two different energy regions.

**Figure 3 nanomaterials-12-02002-f003:**
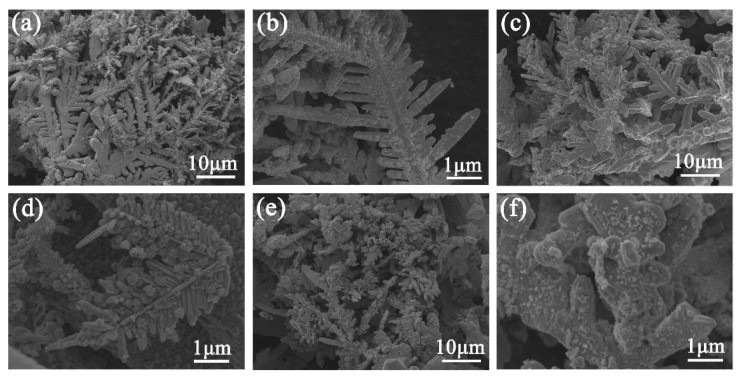
SEM images of 3D Au–Ag substrate obtained at different reaction times: (**a**,**b**) 5, (**c**,**d**) 15 and (**e**,**f**) 30 min.

**Figure 4 nanomaterials-12-02002-f004:**
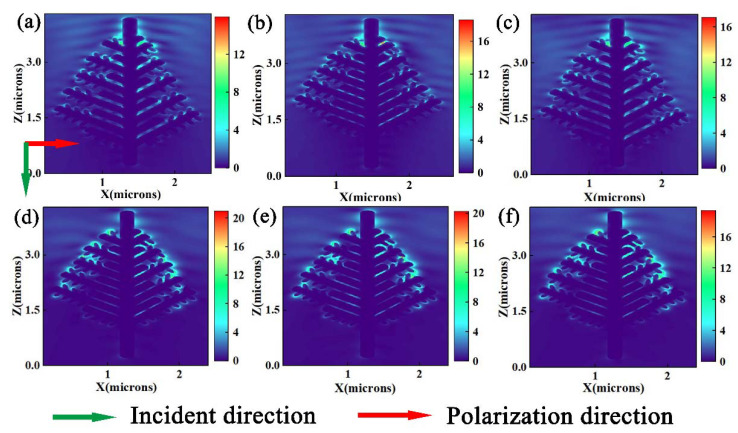
FDTD simulates the electric field distribution of different substrate structures. 3D Ag substrate: (**a**,**b**) the distances between adjacent branches were 80 nm and 40 nm, respectively. 3D Au–Ag substrate: (**c**,**d**) 60 nm Au particles were constructed on the basis of (**a**,**b**) structures respectively, (**e**,**f**) 90 nm and 120 nm Au particles were constructed on the basis of (**b**) structures.

**Figure 5 nanomaterials-12-02002-f005:**
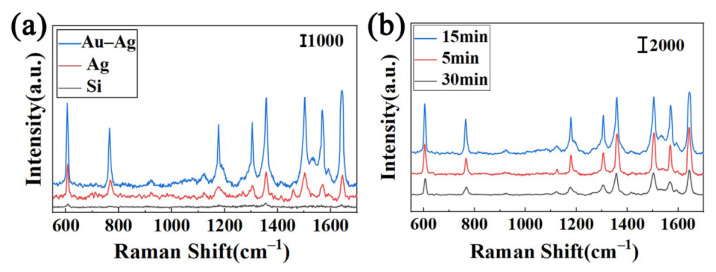
(**a**) SERS spectra of different substrate structures adsorbed to 10^−9^ mol/L R6G solution. (**b**) The Raman spectra of R6G on Au–Ag obtained with different reaction times.

**Figure 6 nanomaterials-12-02002-f006:**
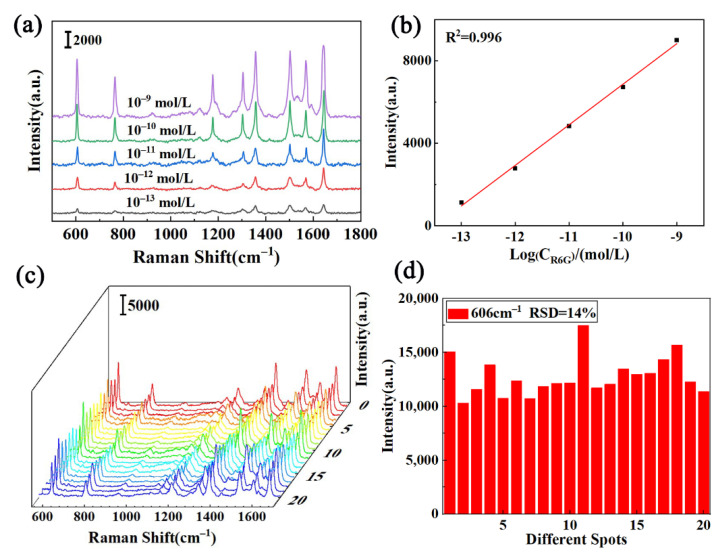
(**a**) SERS spectra of dendritic Au–Ag nanoparticles adsorbed to different concentrations of R6G. (**b**) Plot of intensity at 1645 cm^−1^ of R6G with different concentrations. (**c**) Raman spectra of the 10^−7^ mol/L R6G and (**d**) the corresponding SERS intensity at 606 cm^−1^, which were obtained from 20 different random locations of the Au–Ag substrate.

**Figure 7 nanomaterials-12-02002-f007:**
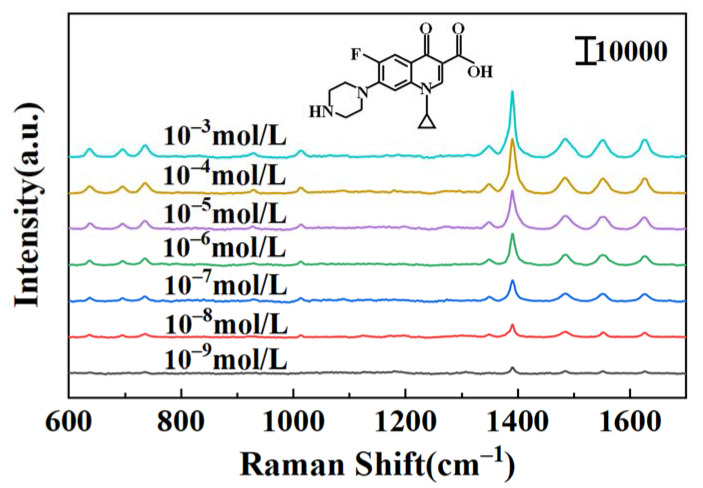
SERS spectra of dendritic Au–Ag nanoparticles adsorbed to different concentrations of ciprofloxacin.

## Data Availability

Not applicable.

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
