# Peer review of "Three-Dimensional Dendritic Au–Ag Substrate for On-Site SERS Detection of Trace Molecules in Liquid Phase"

_nanomaterials, 2022, doi:10.3390/nano12122002_

Round 1
Reviewer 1 Report
see attachment

Reviewer 2 Report
In this work, Authors present a three-dimensional dendritic Au-Ag nanostructure for SERS application, that was constructed by two-step electro displacement reaction in a capillary tube for the on-site liquid phase detection of trace molecules. Using R6G solution as a model molecule, it was shown that the SERS substrate exhibited a detection sensitivity (10-13 mol/L). Authors claim, that the sensor could successfully realize rapid on-site SERS detection of trace molecules in liquids, which provided a promising platform of ultra-sensitive and on-site liquid sample analysis. This work is relevant and interesting and can be published in this journal, but I have a few comments:
1) Authors use in their work Au-Ag metal. But Ag can relatively rapidly oxidize / sulfide. How stable is your substrate?
2) How was the direction of polarization in the FDTD simulation presented in Fig. 4. It shouts to be indicated. Does the direction of polarization affect the amplification efficiency?
Reviewer 3 Report
The proposed manuscript is dedicated to the preparation of a three-dimensional dendritic Au-Ag substrate. The presented approach is noteworthy intriguing and deserves attention from the broad auditorium of the Nanomaterials Journal. However, the design of experiments is questionable, and in general, the manuscript contains a lack of experimental details and the required information. Therefore, the manuscript should be carefully revised before acceptance.
1. First, the characterization of the prepared material is not sufficient. At least, the EDX mapping and XPS spectra should be provided in order to check the distribution of gold and the chemical state of metals. Both factors can affect the performance of SERS. Also, UV-vis spectra should be provided.
2. The authors varied the conditions of preparation. Which sample has been used for the SERS measurements? It will be reasonable to compare the SERS performance for the different structures.
3. The probe treatment should be described in a more precise way. How was the sensor treated after interaction with the solution of the analyte? Just drying?
4. The peak assignation for both analytes should be provided in SI.
5. The R6G demonstrates considerable fluorescence at 532 nm. How did the authors proceed the spectral data?
6. The LOD should be calculated according to common methods.
7. Analytical performance should be analyzed with more care. The authors should check the reproducibility between samples and across one sample,, as well as the stability of sensors in storage in air.
8. Comparison of the prepared sensors with other microfluidic SERS devices should be provided.
Reviewer 4 Report
The author presents Au-Ag dendrites for on-Site SERS Detection of Trace Molecules in Liquid Phase. The work is interesting, and the results are systematically presented. However, there need a few improvements before the publication.
Comments:
1. The author should calculate the enhancement factor.
2. Reproducibility, Repeatability, and Stability of the SERS substrate should be performed.
3. It would be better to add the EDS mapping image instead of the EDS spectrum.
4. There is no comparison of the SERS performance of different SERS substrates obtained at different reaction times.
5. I suggest comparing a few recent Au, Ag, and AuAg-based SERS substrates in the introduction section.
Nanomaterials 12, no. 3 (2022): 401 Journal of Alloys and Compounds 868 (2021): 159136., Journal of Alloys and Compounds, 888, p.161504.
Round 2
Reviewer 3 Report
The proposed manuscript can be accepted.
Reviewer 4 Report
The revised version of the manuscript is acceptable for publication.